# Surrogate Trust in the Intensive Care Unit: What Can We Do Better?

**DOI:** 10.3390/healthcare13101189

**Published:** 2025-05-20

**Authors:** Abdul W. Kazi, Phoebe Chun, David Oxman, Erika J. Yoo

**Affiliations:** 1Department of Medicine, Thomas Jefferson University, Philadelphia, PA 19107, USA; phoebe.chun@jefferson.edu (P.C.); david.oxman@jefferson.edu (D.O.); erika.yoo@jefferson.edu (E.J.Y.); 2Division of Pulmonary, Allergy & Critical Care, Jane and Leonard Korman Respiratory Institute, Thomas Jefferson University, Philadelphia, PA 19107, USA

**Keywords:** trust, intensive care unit, surrogate

## Abstract

Background/Objectives: Previous investigation into patients’ trust in medical providers is largely limited to the outpatient setting, where increased trust is associated with improved adherence, satisfaction, and self-reported quality of life. Contrastingly, in the intensive care unit (ICU), patients are frequently incapacitated, and it is surrogates that engage with clinicians for decision making. This pilot study aims to measure and compare surrogate trust in the healthcare system and ICU physicians. Methods: We measured surrogate trust in both the healthcare system and providers by administering two domain-specific trust-measuring surveys (the Healthcare System Distrust Scale and a modified Trust in Physicians Scale) to surrogates of mechanically ventilated medical ICU patients at an urban academic medical center between November 2021 and April 2024. Results: Responses from twenty-seven surrogates were included in the analysis. The overall mean distrust in healthcare system score was 19.29 (SD 5.8). The overall mean trust in individual physician score was 42.4 (SD 6.26). When subdivided into specific domains, surrogates reported higher mean scores for trust in healthcare system and physician competence than values. Conclusions: In our population, surrogates for medical ICU patients are overall more trusting of their medical providers than the healthcare system. Moreover, surrogates are more trusting of a provider’s professional competence and knowledge than a provider’s values. Our findings may inform trust-building interventions designed specifically for the high-acuity ICU setting to improve quality in communication and family-centered care.

## 1. Introduction

There are many definitions of trust in the literature, one of which is the “optimistic acceptance of a vulnerable situation in which the ‘truster’ believes the ‘trustee’ will care for their interest” [1]. This concept has significant impact for clinicians practicing in the intensive care unit (ICU), where establishing trust with patients or their surrogates can be challenging due to illness acuity and complexity. Previous investigation has shown that longitudinal relationships with patients can foster improved trust and communication [2], a challenging proposition in the ICU where providers often meet patients and their surrogates for the first time. The complexity of critical illness may also necessitate multiple specialists, which can further exacerbate miscommunication and impair trust building.

Factors associated with patient distrust of ICU clinicians are varied and range from race, gender, health history, patient education level, physician communication skills, and even physician attire [3,4,5,6,7,8,9]. Measuring and identifying factors influencing trust is challenging, but lack of trust has a powerful impact. It can exacerbate provider moral injury by augmenting feelings of being unable to provide the highest quality care [10]. It can also increase the risk of burnout [11] or cause conflict regarding non-beneficial care at the end of patient life [12]. Lack of trust is also associated with a lower participation rate in clinical trials [13]. In the ICU, surrogates have a crucial role in medical decision making as many patients are incapacitated, with up to 95% of critically ill adults unable to make their own medical decisions [14]. Previous investigation shows that surrogates have different expectations of trust from different members of the healthcare team [15] (e.g., physicians vs. nurses) and that their expectations of trust can be affected by physician behavior [16].

In light of the limited insight into surrogate trust in the ICU setting, we aimed to quantitively explore surrogate trust both in providers and the healthcare system over the course of a patient’s critical illness and assess the feasibility of survey-based trust measurement within our urban tertiary ICU. In addition to describing provider or patient demographic factors, we measured surrogates’ trust in both physicians and the healthcare system and further subdivided those measurements into various domains assessing competence (provider knowledge or ability) and shared values (beliefs regarding what is important).

## 2. Materials and Methods

This pilot prospective cohort study was conducted in a 17-bed medical ICU of a single tertiary academic center located in Philadelphia, Pennsylvania, United States of America. Critical care in this unit is provided under a high-intensity (closed ICU) model led by board-certified intensivists [17]. Medical care teams in this unit are composed of Internal Medicine house staff supplemented by advanced practice clinicians. Daily rounds are conducted in a multidisciplinary fashion, including bedside nurses, pharmacists, respiratory therapists, dieticians, and physical therapists, as appropriate. Care protocols such as ventilator weaning, progressive mobility, and sedation interruption are enacted per institutional standard.

Screening for eligibility was conducted by study staff. Patients aged >18, mechanically ventilated, and incapacitated with an identified surrogate were eligible for inclusion. Surrogates were identified by the primary medical team’s assessment and included both next of kin and power of attorney (POA). Eligible patients were screened via the electronic medical record (EMR) by key study personnel during screening periods when study personnel were not directly involved in the patient’s care. Surveys were only administered to the primary surrogate decision maker and did not include other family members. Enrollment and informed consent of the surrogate were initially conducted within 48 h of the patient’s ICU admission via in-person discussion whenever possible. If the surrogate was not present in the ICU, consent was obtained via telephone. To optimize enrollment opportunities, the protocol was amended mid-recruitment to extend enrollment to 72 h.

We excluded patients with non-English-speaking surrogates, those expected to have death or discharge within 48 h of admission, and those on chronic mechanical ventilation. Surrogates provided informed consent as all involved patients were incapacitated at the time of enrollment. In the event of the patient’s return of capacity due to medical improvement, the patient was consented for the study. All patients able to provide consent after data collection did so.

After consent was obtained, surrogates were surveyed on days 1−3 of ICU admission with the Healthcare System Distrust Scale and a modified Trust in Physicians Scale [18,19]. A printed copy of each survey was given to the surrogate by study staff for completion. Completed surveys were returned to study personnel for compilation of responses into a central repository. These surveys have been previously used and validated in studies measuring trust in the ICU, and they have been amended to exclude questions related to outpatient medical care [20]. We did not re-test the reliability or validity of these survey tools. Provider (identified as the attending physician of record) demographic information was also collected at the time of survey administration. Data collection ceased upon patient death or discharge from the ICU. Patient, surrogate, and provider demographic characteristics were compiled and managed in a password-protected data repository, REDCap (Version 15.3.3, Vanderbilt University, Nashville, TN, USA) [21,22]. Microsoft Excel (Version 2504, Microsoft Corporation, Redmond, United States) software was used for comparative statistics. This study was approved by the Jefferson Office of Human Research Protection (21D.673).

We measured healthcare system distrust among surrogates using the nine-item revised Healthcare System Distrust Scale. This scale has been used in previous investigations measuring distrust in the ICU setting [20] and has had the wording amended to be applicable to the surrogate of a hospitalized patient. Each item is answered on a 5-point Likert scale (1 = strongly disagree, 5 = strongly agree), and individual items of the scale can be grouped into subscales assessing distrust in technical competence or value congruence. The minimum score for the full scale is 9, and for both subscales, it is 5. The maximum score for the full scale is 45 (25 for the values subscale, and 20 for the competence subscale). Higher scores indicate more distrust.

An adapted version of the Trust in Physicians Scale (TIPS) was also administered to surrogates [17,20]. This adapted ten-item survey assesses trust specifically in the ICU physician and was amended to exclude questions related to outpatient medical care. Each item is answered on a 5-point Likert scale (1 = strongly disagree, 5 = strongly agree), with higher scores indicating more trust. Individual items of the scale can be grouped into subdomains assessing competence or value-based subdomains such as honesty, confidentiality, and compassion.

## 3. Results

A total of 83 eligible participants (surrogates) were approached for enrollment between November 2021 and April 2024, and 27 (32.5%) consented to participate in the study. Patient and, when relevant, surrogate characteristics are presented in Table 1. Both patients and surrogates were more commonly female and of Black or White race. Most patients were admitted to the ICU through the emergency department. Patients’ primary residences spanned 22 zip codes, with 8 being from Philadelphia, Pennsylvania. The remainder of the patients hailed from the surrounding region, including New Jersey (five zip codes) and Delaware (two zip codes), with the furthest zip code being approximately 90 miles south of Philadelphia.

The distribution of the surrogates’ responses to the Healthcare System Distrust Scale is shown in Table 2. Responses ranged from a total distrust sum of 9 to 33 (higher scores indicating more distrust), with a mean of 19.29 (SD 5.8). When comparing scores within the domains of values vs. competence, there was more distrust in healthcare system values than competence (mean 2.25 vs. 2.00, respectively). Mean total distrust scores did not differ by sex of surrogate (19.5, SD 6.9; *n* = 9 for male surrogates vs. 19.3, SD 5.39; *n* = 18 for female surrogates, *p* = 0.38). Black surrogates (19.3, SD 4.7; *n* = 11) reported slightly higher healthcare system distrust scores than White surrogates (18.9, SD 6.4; *n* = 12, *p* = 0.72). Mean distrust scores for patients transferred from outside hospitals were higher (20.9, SD 6.42; *n* = 8) than for patients who were not transferred (18.7, SD 5.4; *n* = 19, *p* = 0.48).

The results of the TIPS are shown in Table 3, with responses ranging from a total trust sum of 22 to 50, with an overall mean of 42.4 (SD 6.26), with higher scores indicating more trust. Mean trust in provider competence scored higher than provider values (4.41 vs. 4.16, respectively). Male surrogates (42.5, SD 5.3; *n* = 9) reported similar trust levels as compared to female surrogates (42.2, SD 6.83; *n* = 18, *p* = 0.77). Black surrogates (42.5, SD 5.3; *n* = 11) were similarly trusting of their physicians as compared to White surrogates (42.2; SD 6.83; *n* = 12, *p* = 0.90). The surrogates of patients transferred from an outside hospital (39.3; SD 8.3; *n* = 8) were less trusting than surrogates whose loved ones were never transferred (43.6, SD 4.9; *n* = 19, *p* = 0.19).

## 4. Discussion

In our pilot study quantitively measuring trust among surrogates for critically ill patients, we found that surrogates are more trusting of individual physicians than the healthcare system, which is consistent with prior investigations [20,23]. Furthermore, we identified that surrogates are more trusting of provider competence (i.e., technical knowledge and skills) than provider values (honesty, compassion, etc.). Trust levels may also be affected by other factors, such as interhospital transfer, as healthcare system distrust was higher, and physician trust was lower amongst surrogates whose loved ones underwent transfer.

Our findings of higher levels of distrust in the healthcare system are consistent with the prior literature [20]. Such institutional distrust is also common in the legal and political system [24,25] and may be representative of a broader societal shift toward distrusting expert institutions [26]. Surrogates may experience high healthcare costs, insurance complexities, and perceptions that healthcare decisions are driven by financial motivations rather than patient-centered motivations [27]. All such factors may worsen satisfaction with and trust in the healthcare system.

Despite broader trends of institutional and healthcare system distrust [26], surrogates are relatively more trusting of their loved ones’ individual physicians. This may be due to surrogates being able to put a “face” to their physician-expert as opposed to a healthcare system in which competence and values are more opaque. Interacting with surrogates directly allows physicians to convey their technical skills, clinical acumen, experience, leadership, empathy, reliability, and commitment to confidentiality. This may be either through direct communication or non-verbal aspects of the physician–surrogate interaction which have been shown to affect patient-centered outcomes such as satisfaction [28]. Higher levels of trust in ICU physicians may be driven more by surrogates’ reassurance of a physician’s knowledge rather than values. The complexity of ICU patients is highly visible to surrogates. For example, the need for invasive proceduralization and diagnostic tests or the intricacy and multiplicity of life-sustaining medications and machines may act as indirect indicators of high provider competence. However, there is no such evident indicator for provider values that would help signal to surrogates that providers care about the same things that they do.

If surrogates are inherently more trusting of their doctor’s competence rather than their values, then clinician focus should be directed towards behaviors which improve value-based trust rather than competence-based trust. This can be accomplished in a multitude of ways, such as utilization of communication tools [9] that emphasize what the patient and the surrogate care about. These tools can provide a communication framework which emphasizes clear communication which is honest, inclusive, and compassionate. Family meetings can provide emotional support and elicit patient values, and they have been shown to increase trust in providers [16], especially when meetings incorporate multiple elements of shared decision making. Such interactions provide an opportunity for providers to learn a patient’s trust-relevant characteristics (family structure, religion, cultural background, etc.) and inquire about a patient’s (or surrogate’s) values. Providers can focus on modifying factors within their control to build on shared values (e.g., honesty in disclosing mistakes) rather than to demonstrate technical or clinical skill [29]. The absence of value-based trust may be more impactful on a patient–provider relationship due to negative emotional effects from perceived failures in integrity. This may come at a high inter-relationship cost due to the lack of therapeutic alliance [30].

This pilot study had several limitations. Firstly, there were difficulties in recruitment and timely survey administration due to the limited time frame allowed for enrollment after ICU admission. Although the survey protocol was amended to allow for a longer enrollment window, the sample size was consequently small. There is also the potential for selection bias, as surrogates who are inherently more trusting may be more likely to consent to participation in a study about trust. Study generalizability may also be impacted by convenience sampling, but our study sample exhibits demographic heterogeneity, which may help to ameliorate this bias. Because of the small sample size, we were unable to make definitive conclusions about contextual factors that may influence surrogate trust (e.g., interfacility transfer). However, our observations can be viewed as hypothesis-generating and may warrant more granular investigation.

Despite these challenges, the overall purpose of this pilot study was to validate previously used tools in our local population and create a quantifiable exploratory framework upon which further studies may be designed. These future directions can include the implementation of trust-building interventions over various platforms such as family-centered rounds, family meetings, or periodic telephone or in-person updates [31]. There may also be opportunity for investigation into disparities in trust among patient populations based on differences in race, exposure to palliative care, or experience of interfacility transfer. Future study design may be supplemented by methodological improvements such as a multi-center approach, more longitudinal data collection, or the addition of qualitative assessments of trust.

## 5. Conclusions

Trust is an essential component of the physician–patient relationship and has been shown to improve various clinical outcomes. But trust between patients’ surrogates and physicians, which is likely more relevant in the critical care setting, is less understood. This pilot study quantitively explored surrogate trust within the ICU by administering previously validated trust-measuring survey instruments to surrogates within a demographically diverse population at our urban tertiary care institution. The results showed relatively higher levels of trust in physicians compared to the healthcare system. Furthermore, when divided into specific subdomains, surrogates exhibited more trust in physician competence than values. These findings highlight the need for investigation into specific factors contributing to such differences in trust levels in the ICU. These factors may be demographic, clinical, provider-based, or environmental. Further study aimed at identifying potential interventions to increase value-based trust will also support the patient- and family-centered care research agenda in the ICU [32].

## Figures and Tables

**Table 1 healthcare-13-01189-t001:** Patient and surrogate characteristics *.

Sex	**Patient (*n* = 27)**	**Surrogate (*n* = 27)**
Female	15 (55)	18 (66)
Male	12 (44)	9 (33)
Age (y)	60 (7.3)	
Surrogate relationship to patient		
Spouse	4 (15)	
Child	8 (30)	
Parent	6 (22)	
Other	5 (19)	
Admission source		
Emergency department	12 (44)	
Another hospital	6 (30)	
Ward/floor	5 (26)	
Race	**Patient (*n* = 27)**	**Surrogate (*n* = 27)**
White	12 (44)	12 (44)
Black	11 (41)	11 (41)
Asian	3 (11)	3 (11)
Hispanic	1 (4)	1 (4)
COVID-vaccinated	23 (85)	
PCP visit within last 3 years	22 (81)	
Smoking status		
Current	7 (26)	
Former	7 (26)	
Nonsmoker	13 (48)	
Marital status		
Married	7 (26)	
Divorced	4 (15)	
Single	12 (44)	
Widowed	4 (15)	
History of substance use (alcohol, IVDU)	12 (44)	
Median income of zip code of residence		
0−50k	6 (22)	
50k−100k	12 (44)	
150−200k	9 (33)	
200k+	0	

* Data presented as *n* (%) and mean (SD) where appropriate. PCP = primary care provider; IVDU = intravenous drug use.

**Table 2 healthcare-13-01189-t002:** Healthcare System Distrust Scale results *.

Item	Domain	Mean (SD)
The Health Care System does its best to make patients’ health better ^†^	Competence	1.80 (0.92)
The Health Care System covers up its mistakes	Values	2.44 (0.89)
Patients receive high quality medical care from the Health Care System	Competence	2.00 (1.07)
The Health Care System makes too many mistakes	Competence	2.25 (0.71)
The Health Care System puts making money above patients’ needs	Values	2.51 (1.25)
The Health Care System gives excellent medical care	Competence	1.96 (0.89)
Patients get the same medical treatment from the Health Care System, no matter what the patient’s race or ethnicity	Values	2.25 (1.05)
The Health Care System lies to make money	Values	2.18 (1.07)
The Health Care System experiments on patients without them knowing	Values	1.96 (0.89)
Distrust in Values		2.25
Distrust in Competence		2.00
Overall Distrust		19.29

* Items are answered on a scale where 1 = strongly disagree and 5 = strongly agree, with the total possible score ranging from 9 to 45. ^†^ Inverted item.

**Table 3 healthcare-13-01189-t003:** Trust in Physicians Scale *.

Item	Domain	Mean (SD)
I doubt that ____’s doctors really care about him/her as a person	Dependability	4.44 (0.89)
____’s doctors are usually considerate of his/her needs and puts them first *	Dependability	4.37 (0.79)
If ____’s doctors tell me something is so, then it must be true	Dependability	4.14 (0.91)
I sometimes distrust ____’s doctors’ opinions and would like a second one	Knowledge/Skills	4.07 (0.98)
I trust ____’s doctors’ judgments about his/her medical care *	Knowledge/Skills	4.41 (0.80)
I feel ____’s doctors do not do everything they should about his/her medical care	Dependability	4.04 (1.26)
I trust ____’s doctors to put his/her medical needs above all other considerations when treating his/her medical problems *	Dependability	4.33 (0.88)
____’s doctors are well qualified to manage (diagnose and treat or make an appropriate referral) medical problems like his/hers	Knowledge/Skills	4.67 (0.62)
I trust ____’s doctors to tell me if a mistake was made about his/her treatment	Dependability	3.81 (0.96)
I sometimes worry that ____’s doctors may not keep the information we discuss totally private ^†^	Confidentiality	4.04 (1.22)
Trust in Competence (Knowledge/Skills)		4.41
Trust in Values (Dependability, Confidentiality)		4.16
Overall Trust in Physicians		42.4

* Items are answered on a scale where 1 = strongly disagree and 5 = strongly agree, with the total possible score ranging from 10 to 50. ^†^ Inverted item.

## Data Availability

Deidentified survey response data presented in this study are available on request from the corresponding author.

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
