# Peer review of "Surrogate Trust in the Intensive Care Unit: What Can We Do Better?"

_healthcare, 2025, doi:10.3390/healthcare13101189_

Round 1

Reviewer 1 Report

Comments and Suggestions for Authors

Dear Authors,
This article not only thoroughly examines the topic [Surrogate trust in the intensive care unit: Where can we do better?], but also adds to our knowledge and understanding of this field by providing detailed analyses and significant results. Your efforts and meticulousness in researching and writing this article are truly commendable and will have a significant impact on the scientific community. Once again, I thank you for your efforts and hope to see more valuable works from you in the future.
The following topics are suggested for improving the article
The sampling method should be stated more precisely
The discussion can be better and more precise based on the findings

General Comments

The present article is well-explained, but a few suggestions can be made to enhance its clarity and depth.

Specific Comments

Methods Section

To improve the clarity of the methods section, the following division is suggested:

  1. Introduction of the Tool: Include discussions on the reliability and validity of the tools used.
  2. The Study Population: Clearly define the demographic characteristics of the participants.
  3. Inclusion and Exclusion Criteria: Specify the criteria used to select participants and exclude others.
  4. How to Complete the Questionnaire: Provide detailed instructions on how the questionnaire was administered and completed?

Results Section

A better explanation of the findings would be beneficial. A more detailed analysis of the results can help readers understand the issue more comprehensively.

Discussion Section

To strengthen the discussion, the first paragraph on page 6 should be supported with relevant references. Additionally, further development of this section can enhance the overall explanation of the topic.

These suggestions aim to improve the manuscript's clarity and depth, ensuring it aligns well with the journal's standards.

Thank you for considering these comments.

Author Response

Methods Section

To improve the clarity of the methods section, the following revision is suggested:

Comment 1: Introduction of the Tool: Include discussions on the reliability and validity of the tools used.

Response 1: Further descriptions regarding the origin, reliability, and validity of the survey tools has been added to the Methods section. The Trust in Physician (TIPS) and Healthcare System Distrust Scales (HCDS) have been previously validated in the critical care setting and the relevant citation is now included in the Methods section.

Comment 2: The Study Population: Clearly define the demographic characteristics of the participants.

Response 2: Description and demographic characteristics of the study population are included in Results Table 1 and within the Results section.

Comment 3: Inclusion and Exclusion Criteria: Specify the criteria used to select participants and exclude others.

Response 3: Further elucidation of inclusion criteria and exclusion criteria were added to the Methods section.

Comment 4: How to Complete the Questionnaire: Provide detailed instructions on how the questionnaire was administered and completed?

Response 4: Instructions provided to study participants per study protocol were added to this section. This includes information regarding delivery of survey, medium, and subsequent compilation of results.

Results Section

Comment 5: A better explanation of the findings would be beneficial. A more detailed analysis of the results can help readers understand the issue more comprehensively.

Response 5: We agree with the Reviewer that further explanation and contextualization of these results would bring clarity to the reader. We have modified the Discussion section of the paper to provide a more detailed analysis, reserving this section for presentation of results only. 

Discussion Section

Comment 6: To strengthen the discussion, the first paragraph on page 6 should be supported with relevant references. Additionally, further development of this section can enhance the overall explanation of the topic.

Response 6: Addition of an appropriate reference was added to the first paragraph of page 6, thereby strengthening the opening sentence.  This paragraph was further developed by including relevant references which may explain the mechanisms of how providers display competence-based trust (ie, through face-to-face non-verbal communication). 

Reviewer 2 Report

Comments and Suggestions for Authors

The topic of the manuscript is very interesting and may be of importance for further development of theoretical models and improvement of practice. It is evident that the authors paid a lot of attention to details and thus achieved a high academic level in writing. There are some details that deserve special attention.

1. The authors should consider a small correction of the title. The phrase "What can we do better?" is more appropriate in the context of this paper because the primary focus is on identifying specific problems and proposing actionable solutions to improve surrogate trust in the intensive care unit. While "Where can we do better?" suggests a geographical or contextual exploration of weak points, "What can we do better?" directly addresses the need for practical interventions and changes in practice. Since this paper aims to analyze concrete issues and provide recommendations, the use of "what" aligns better with the purpose and content of the work.

2. The Introduction section is well structured and informative, and at the end of this section the research objective is clearly defined.

3. The Materials and Methods section is probably the best part of the paper. All necessary elements are described clearly and in detail.

4. The displayed results are clearly presented in tables. Despite this, it is important to state that the sample is very small and that the research lacks representativeness. The authors clearly addressed this as a limitation of the study and stated that the current research is a pilot study.

The given tables present a large number of categorical variables (Sex, Surrogate relationship to patient, Race, Smoking status, etc.). This is excellently done, but it is unclear why the impact of these factors on the perception of trust was not tested and analyzed. By comparing attitudes between different groups, results could be obtained that will indicate clearer guidelines for future research, and at the same time, important practical implications would be determined. The authors are recommended to do additional analyzes and present these results.

5. Probably the biggest shortcoming of the study is the neglect of the influence of the person-to-person relationship, including various sociological aspects, which undoubtedly have an effect on the perception of trust. Essentially, the study does not address the causes of why the current state of trust is as it is, but only analyzes the consequences. They are certainly important, but have significantly less analytical value. The authors partially addressed this in the limitations section of the manuscript, but more attention should be paid to this aspect in the discussion itself, and perhaps in the Introduction section.

6. The manuscript lacks a Conclusion section.

Author Response

Comment 1: The authors should consider a small correction of the title. The phrase "What can we do better?" is more appropriate in the context of this paper because the primary focus is on identifying specific problems and proposing actionable solutions to improve surrogate trust in the intensive care unit. While "Where can we do better?" suggests a geographical or contextual exploration of weak points, "What can we do better?" directly addresses the need for practical interventions and changes in practice. Since this paper aims to analyze concrete issues and provide recommendations, the use of "what" aligns better with the purpose and content of the work.

Response 1: Thank you for this suggestion. The title has been updated as requested. 

Comment 2: The Introduction section is well structured and informative, and at the end of this section the research objective is clearly defined.

Response 2: Thank you for this comment. The Introduction was minimally updated in response to another Reviewer’s comments. However, the overall clarity of the section remains.

Comment 3: The Materials and Methods section is probably the best part of the paper. All necessary elements are described clearly and in detail.

Response 3: Thank you for this comment. Further clarification of study protocol was provided at the request of another Reviewer.

Comment 4: The displayed results are clearly presented in tables. Despite this, it is important to state that the sample is very small and that the research lacks representativeness. The authors clearly addressed this as a limitation of the study and stated that the current research is a pilot study. The given tables present a large number of categorical variables (Sex, Surrogate relationship to patient, Race, Smoking status, etc.). This is excellently done, but it is unclear why the impact of these factors on the perception of trust was not tested and analyzed. By comparing attitudes between different groups, results could be obtained that will indicate clearer guidelines for future research, and at the same time, important practical implications would be determined. The authors are recommended to do additional analyses and present these results.

Response 4: As a pilot study, we aimed to assess the feasibility of these research instruments at our institution to quantitatively measure trust as a construct within our study population. As such, the results displayed within Table 2 and 3 are predominantly descriptive. As the Reviewer notes, with an admittedly small sample size, it is difficult to make associations between patient or surrogate characteristics and individual trust scores. However, we do agree with the Reviewer that within the body of the Results section there is opportunity for the addition of statistical comparisons, which are now included. The Methods were updated to reflect these additions.

Comment 5: Probably the biggest shortcoming of the study is the neglect of the influence of the person-to-person relationship, including various sociological aspects, which undoubtedly have an effect on the perception of trust. Essentially, the study does not address the causes of why the current state of trust is as it is, but only analyzes the consequences. They are certainly important, but have significantly less analytical value. The authors partially addressed this in the limitations section of the manuscript, but more attention should be paid to this aspect in the discussion itself, and perhaps in the Introduction section.

Response 5: We agree with the Reviewer that the purpose of this study, to assess the feasibility of measuring trust as a construct in our ICU, does not address risk factors for trust or mistrust. As a pilot study validating the instruments and reporting descriptive findings, these results establish groundwork which can be used to design and implement further research that answer such questions. This is now more clearly stated in the Discussion.  

Comment 6: The manuscript lacks a Conclusion section

Response 6: We appreciate the Reviewer’s recommendation for a Conclusion section and have added this as requested.

Reviewer 3 Report

Comments and Suggestions for Authors

Abstract

Clarify the terms "values" and "competence" as they might be unclear to readers unfamiliar with the scales used.

Consider stating clearly the significance or potential impact of observed differences in trust scores (e.g., practical implications).

Replace awkward phrasing

Ensure consistency in formatting and hyphenation.

Introduction

More explicitly define how your study builds upon or diverges from cited studies.

Provide clearer transitions between ideas (e.g., from general definitions of trust to specific ICU-related challenges).

Provide stronger rationale for the focus on surrogates (e.g., frequency or impact of surrogate decision-making in ICUs).

Clarify the sentence: "It can exacerbate provider moral injury or cause conflict regarding non-beneficial care…" Consider elaborating briefly on "moral injury."

Avoid using overly generalized phrases like "even less is known." Quantify or specifically highlight knowledge gaps more concretely.

Materials and Methods

Clearly state how "modified" scales differ from original versions and justify why modifications were necessary.

Clarify why surrogates were approached via phone/in-person and how potential biases (e.g., recruitment methods, convenience) were mitigated.

Explicitly mention whether reliability/validity of modified instruments was tested within your study.

Results

Consider adding inferential statistics (e.g., t-tests, ANOVA) to determine if differences observed (such as between transferred vs. non-transferred patients) are statistically significant, even with the small sample.

Reformat tables slightly for readability (e.g., bold significant differences, clarify table notes).

Ensure consistency in decimal rounding across tables.

Avoid merely reporting numerical scores; briefly contextualize the meaning (e.g., interpreting what a mean distrust of ~19 implies practically or clinically).

Discussion

Go deeper in hypothesizing why surrogates have lower trust in healthcare systems versus individual providers. Consider explicitly referencing social or systemic factors more clearly.

Explain more explicitly the clinical implications of higher competence-based trust vs. lower values-based trust (e.g., how this influences decision-making processes or surrogate satisfaction).

Offer clearer, actionable recommendations or examples of trust-building interventions suitable for ICU staff to improve "value-based" trust.

Clearly articulate how recruitment challenges specifically might have skewed results or affected generalizability.

Suggest specific methodological improvements (e.g., multi-center approach, qualitative elements, longitudinal design).

Conclusion/Overall Implications

Clarify how the pilot nature of the study impacts conclusions, and avoid over-generalizing results.

Suggest explicitly how these findings might immediately influence clinician-surrogate interactions or healthcare policies in ICU settings.

Comments on the Quality of English Language

Consider rephrasing longer, complex sentences for clarity.

Verify consistent use of hyphenation.

Double-check the numerical consistency and formatting of statistics throughout the text.

Several minor grammatical improvements are needed.

Use clear phrasing: “due to illness acuity and complexity” → consider specifying "acute illness severity and clinical complexity."

Author Response

Comment 1: Clarify the terms "values" and "competence" as they might be unclear to readers unfamiliar with the scales used.

Response 1: Definitions for ‘values’ and ‘competence’ were added within the Introduction section.

Comment 2: Consider stating clearly the significance or potential impact of observed differences in trust scores (e.g., practical implications).

Response 2: The overall impact of low levels of trust in a physician-patient/surrogate relationship is summarized in the Introduction with appropriate references. As requested, further contextualization of the significance of changes in trust levels was included within the Discussion section as well.

Comment 3: Replace awkward phrasing. Ensure consistency in formatting and hyphenation.

Response 3: We have re-reviewed the diction, syntax, and formatting of the manuscript and amended any awkward phrasing or inconsistencies.

Introduction

Comment 4: More explicitly define how your study builds upon or diverges from cited studies.

Response 4: The purpose of this pilot study is to quantitively explore surrogate trust within our patient population by using previously validated trust scales. The relationship of our study to previous research is elucidated in the Discussion.

Comment 5: Provide clearer transitions between ideas (e.g., from general definitions of trust to specific ICU-related challenges).

Response 5: Transitions within the Introduction paragraph were improved as requested.

Comment 6: Provide stronger rationale for the focus on surrogates (e.g., frequency or impact of surrogate decision-making in ICUs).

Response 6: In the critical care setting, where patients are often incapacitated due to critical illness, the surrogate is their sole decision maker. We updated study rationale in the Introduction with the addition of appropriate references.

Comment 7: Clarify the sentence: "It can exacerbate provider moral injury or cause conflict regarding non-beneficial care…" Consider elaborating briefly on "moral injury”.

Response 7: A clarification and citation for “moral injury” was provided  where indicated in text.

Comment 8: Avoid using overly generalized phrases like "even less is known." Quantify or specifically highlight knowledge gaps more concretely.

Response 8: As investigation of trust specifically in the critical care setting is limited, quantifying knowledge gaps is difficult. However, we have updated the text to avoid use of overly generalized phrases as requested.

Materials and Methods

Comment 9: Clearly state how "modified" scales differ from original versions and justify why modifications were necessary.

Response 9: We clearly stated both the justification and modification of trust scales as requested.

Comment 10: Clarify why surrogates were approached via phone/in-person and how potential biases (e.g., recruitment methods, convenience) were mitigated.

Response 10: Clarification of the protocol regarding approach of surrogates was added to the Methods section. We expand upon potential biases within the Discussion.

Comment 11: Explicitly mention whether reliability/validity of modified instruments was tested within your study.

Response 11: We have updated the text to reflect that the purpose of our study was not to validate these previously validated trust scales.

Results

Comment 12: Consider adding inferential statistics (e.g., t-tests, ANOVA) to determine if differences observed (such as between transferred vs. non-transferred patients) are statistically significant, even with the small sample.

Response 12: In this pilot study, the reported results are descriptive in nature. However, where appropriate within the body of the text, we now report the inferential statistical analysis results, when appropriate. 

Comment 12: Reformat tables slightly for readability (e.g., bold significant differences, clarify table notes). Ensure consistency in decimal rounding across tables.

Response 12: We corrected tables as requested within the confines of author instructions provided by the journal.

Comment 13: Avoid merely reporting numerical scores; briefly contextualize the meaning (e.g., interpreting what a mean distrust of ~19 implies practically or clinically).

Response 13: In the Results, we report numerical scores per standard. In the Discussion, we contextualize the meaning of these scores. We have added clarifications to the Discussion and Conclusion section to address Reviewer’s concerns.

Discussion

Comment 14:  Go deeper in hypothesizing why surrogates have lower trust in healthcare systems versus individual providers. Consider explicitly referencing social or systemic factors more clearly.

Response 14: We have sought to address the Reviewer’s comment in paragraph 3 of the Discussion. We have included additional references to this paragraph to provide context for proposed mechanisms for our findings.

Comment 15: Explain more explicitly the clinical implications of higher competence-based trust vs. lower values-based trust (e.g., how this influences decision-making processes or surrogate satisfaction).

Response 15: We have added the requested explanation to the Discussion and hypothesize its impact with an added citation as appropriate.

Comment 16: Offer clearer, actionable recommendations or examples of trust-building interventions suitable for ICU staff to improve "value-based" trust.

Response 16: We offer actionable recommendations in the Discussion (paragraph 4). We have added to the text of this paragraph for expansion.

Comment 17: Clearly articulate how recruitment challenges specifically might have skewed results or affected generalizability.

Response 17: We better address recruitment challenges, bias, and generalizability in the limitations section of the Discussion.

Comment 18: Suggest specific methodological improvements (e.g., multi-center approach, qualitative elements, longitudinal design).

Response 18: Thank you for this comment. These suggestions were included within the appropriate paragraph of the Discussion section.

Conclusion/Overall Implications

Comment 19: Clarify how the pilot nature of the study impacts conclusions, and avoid over-generalizing results.

Response 19: We have added a Conclusion to the revised manuscript. This section serves to summarize the findings without drawing conclusions that are unsubstantiated.

Comment 20: Suggest explicitly how these findings might immediately influence clinician-surrogate interactions or healthcare policies in ICU settings.

Response 20: In expanded paragraph 4, we propose areas and mechanisms of trust that clinicians should focus on in their interactions with surrogates in an effort to explicitly address “real-world” applicability.

Round 2

Reviewer 2 Report

Comments and Suggestions for Authors

The authors have added a Conclusion section, but it still seems modest. Probably more effort needs to be added to achieve the expected form.  I think the conclusion is short and should be more informative. 

No additional comments.

Author Response

Comment 1: The authors have added a Conclusion section, but it still seems modest. Probably more effort needs to be added to achieve the expected form.  I think the conclusion is short and should be more informative. 

Reply 1: We thank the reviewer for their comment regarding the Conclusion section. We have included additional back ground details, information regarding the study, and have added an additional citation to support our conclusions. 

Reviewer 3 Report

Comments and Suggestions for Authors

The authors addressed all the suggestions carefully.

I have no more comments.

Author Response

Comment 1: 

The authors addressed all the suggestions carefully.

I have no more comments.

Reply 1: We thank the reviewer for their comments